# Triiron Tetrairon Phosphate (Fe_7_(PO_4_)_6_) Nanomaterials Enhanced Flavonoid Accumulation in Tomato Fruits

**DOI:** 10.3390/nano12081341

**Published:** 2022-04-13

**Authors:** Zhenyu Wang, Xiehui Le, Xuesong Cao, Chuanxi Wang, Feiran Chen, Jing Wang, Yan Feng, Le Yue, Baoshan Xing

**Affiliations:** 1School of Environment and Civil Engineering, Institute of Environmental Processes and Pollution Control, Jiangnan University, Wuxi 214122, China; wang0628@jiangnan.edu.cn (Z.W.); 6191403013@stu.jiangnan.edu.cn (X.L.); caoxuesong@jiangnan.edu.cn (X.C.); wangcx2018@jiangnan.edu.cn (C.W.); chenfeiran@jiangnan.edu.cn (F.C.); wangjing03282022@163.com (J.W.); 15251870663@163.com (Y.F.); 2Jiangsu Engineering Laboratory for Biomass Energy and Carbon Reduction Technology, Wuxi 214122, China; 3Stockbridge School of Agriculture, University of Massachusetts, Amherst, MA 01003, USA; bx@umass.edu

**Keywords:** flavonoids, tomato fruits, Fe_7_(PO_4_)_6_ NMs, transcriptomic, metabonomic

## Abstract

Flavonoids contribute to fruit sensorial and nutritional quality. They are also highly beneficial for human health and can effectively prevent several chronic diseases. There is increasing interest in developing alternative food sources rich in flavonoids, and nano-enabled agriculture provides the prospect for solving this action. In this study, triiron tetrairon phosphate (Fe_7_(PO_4_)_6_) nanomaterials (NMs) were synthesized and amended in soils to enhance flavonoids accumulation in tomato fruits. 50 mg kg^−1^ of Fe_7_(PO_4_)_6_ NMs was the optimal dose based on its outstanding performance on promoting tomato fruit flavonoids accumulation. After entering tomato roots, Fe_7_(PO_4_)_6_ NMs promoted auxin (IAA) level by 70.75 and 164.21% over Fe-EDTA and control, and then up-regulated the expression of genes related to PM H^+^ ATPase, leading to root proton ef-flux at 5.87 pmol cm^−2^ s^−1^ and rhizosphere acidification. More Mg, Fe, and Mn were thus taken up into plants. Subsequently, photosynthate was synthesized, and transported into fruits more rapidly to increase flavonoid synthesis potential. The metabolomic and transcriptomic profile in fruits further revealed that Fe_7_(PO_4_)_6_ NMs regulated sucrose metabolism, shi-kimic acid pathway, phenylalanine synthesis, and finally enhanced flavonoid biosynthesis. This study implies the potential of NMs to improve fruit quality by enhancing flavonoids synthesis and accumulation.

## 1. Introduction

Flavonoids, a kind of C6-C3-C6 skeleton phenolic secondary metabolite, play an important role in food organoleptic properties and human health [1]. For example, flavonoids are able to protect cells from oxidative stress [2], regress tumors [3], regulate diabetes and obesity [4], alleviate inflammation, and enhance the immune system [5]. Bondonno et al. (2019) [6] reported that the habitual intake of flavonoids could decrease cardiovascular and cancer induced mortality and recommend humans to increase the intake of flavonoid-rich foods. However, the flavonoids intake of populations has remained low due to eating habits involving flavonoids-lacking foods. Therefore, there is increasing interest in developing alternative food sources rich in flavonoids [7]. Tomatoes (*Solanum lycopersicum* L.), including a wide variety of processed tomato food products (e.g., ketchup, pasta sauce, and tomato puree), are one of the major vegetables in human diets and are therefore an ideal candidate for improved flavonoids intake [8].

To date, nano-enabled agriculture technology is emerging for crop yield and quality improvement. Recent studies discovered that flavonoids could be enhanced with nanomaterials (NMs) applications. For example, 10 mg/L Se and Cu NMs significantly enhanced the content of flavonoids in tomato fruits by 26.4 and 20.9% as compared with control [9]. The application with 250 mg/L SiO_2_ NMs also significantly increased the flavonoids content in tomato fruits [10]. Given these promising findings and the known importance of flavonoids in tomato fruits, future investigation into the mechanisms of action and optimization of efficacy is warranted.

The synthesis of flavonoids in plants is highly regulated by sucrose, which can act as a signaling molecule to affect flavonoid synthesis related genes expression [11,12,13]. In addition, sucrose could activate phenylpropanoid metabolism, which leads to flavonoid synthesis and upregulate the expression of transcription factors (TFs) like *bHLH1*, *MYB* and *WD40* [14,15]. *bHLH* TFs can positively regulate the synthesis of flavonoids [16]. *MYB* TFs can increase the expression of chalcone synthase, chalcone isomer and other enzyme genes in the flavonoids metabolic process [17]. *WD40* is not considered to have any catalytic activity alone, but they act as a bridge combining *MYB* and *bHLH* to form a complex in regulating flavonoid synthesis [18]. Therefore, enhancing the sucrose accumulation in tomato fruits could effectively increase their flavonoids content. It should be noted that sucrose in tomato fruits mainly comes from photosynthetic carbon fixation [19]. The photosynthetic system requires the participation of various elements (e.g., Fe, Mn, and P). However, these elements present low bioavailability in the soil, and plants usually adopt the strategy of acidifying the rhizosphere to obtain them [20]. Plasma membrane (PM) H^+^-ATPase facilitates the transport of various nutrients, such as nitrate, phosphate and potassium, and is responsible for pumping protons out of the plasma to achieve acidification [21,22]. The overexpression of PM H^+^-ATPase could increase rice yield through simultaneous enhancement of photosynthesis and nutrient uptake [23,24]. Besides, Fe and P play a vital role in plant growth [25,26]. Fe was a key element for 2-oxoglutarate dependent (2-ODD) oxygenases that oxidize the central C ring in flavonoid synthase. Therefore, materials such as composite triiron tetrairon phosphate (Fe_7_(PO_4_)_6_) are of particular interest for application as part of a nano-enabled strategy to elevate flavonoids accumulation in tomato fruits. Interestingly, several studies mentioned that Fe-based NMs could induce the activation of root PM H^+^-ATPase [23,27]. In the present study, we hypothesized that tomato roots exposed to Fe_7_(PO_4_)_6_ NMs in soil could trigger high activity of PM H^+^-ATPase to acidify rhizosphere in order to mobilize nutrient elements and promote photosynthesis, more sucrose would accumulate and induce flavonoids biosynthesis in tomato fruits. The goal of this work was to investigate the mechanisms of enhanced flavonoid accumulation in tomato fruits by the amendment with Fe_7_(PO_4_)_6_ NMs, including: (1) PM H^+^-ATPase activity in tomato seedling roots; (2) nutrient uptake, photosynthesis, and sucrose accumulation during the plant growth; and (3) flavonoid synthesis in tomato fruits by the metabolomic and transcriptomic profiles. This study highlights the potential of NMs to improve human health through elevating flavonoid accumulation in dietetic fruits.

## 2. Materials and Methods

### 2.1. Synthesis and Characterization of Fe_7_(PO_4_)_6_ NMs

The chemicals for NM synthesis were all purchased from Sinopharm Chemical Reagent Co., Ltd, Shanghai, China. Pristine Fe_7_(PO_4_)_6_ NMs were synthesized based on the study of Song et al. (2015) [28] with slight modification. Firstly, 3.45 g NH_4_H_2_PO_4_ was dissolved in 20 mL deionized (DI) water at 25 °C, making a 173 mg L^−1^ NH_4_H_2_PO_4_ solution. In order to produce nanoscale suspended particles, 0.5 g PVP-K30 (Polyvinylpyrrolidone K30) was dissolved in 10 mL DI water, and then mixed with the above NH_4_H_2_PO_4_ solution. Then, 2.025 g FeCl_3_·6H_2_O was dissolved in 20 mL DI water, making a 101 mg L^−1^ ferrous solution. Then, the ferrous solution was added dropwise for 6 s per drop into the PVP-K30 and NH_4_H_2_PO_4_ mixture with a magnetic stirrer until all ferrous solution was consumed. Afterwards, the obtained mixture was centrifuged at 6000 rpm for 10 min and the supernatant was discarded. The precipitate was transferred into a 100 mL Teflon-lined stainless steel autoclave. The autoclave was sealed and heated to 180 °C for 6 h. After taking out and cooling down, the products were separated by centrifugation at 5000 rpm for 5 min and washed by DI water for three times. After dried in vacuum oven at 60 °C for 2 h, Fe_7_(PO_4_)_6_ NMs were obtained. The productivity of synthetic Fe_7_(PO_4_)_6_ NMs was 0.92 g. The chemical reaction is shown below:$${\mathrm{Fe}}^{3+}+{\;\mathrm{PO}}_{4}^{3-}\;\to {\mathrm{FePO}}_{4}\overset{\mathrm{PVP}\;K30,\;\mathrm{hydrothermal}\;\left(180\;℃,\;6\;h\right)}{\to }{\;\mathrm{Fe}}_{7}{\left({\mathrm{PO}}_{4}\right)}_{6}$$

The shape and size of synthesized Fe_7_(PO_4_)_6_ NMs were observed by transmission electron microscopy (TEM, JEM–2100, Nippon electronics Co., Tokyo, Japan). Energy dispersive spectroscopy (EDS) was used for Fe, P and O qualitative analysis. The chemical formula composition was examined by X-ray diffraction (XRD, Bruker AXS, Berlin, Germany) and Jade 5 was performed for peak detection. X-ray photoelectron spectroscopy (XPS, Thermo Fisher ESCALAB 250Xi, Waltham, MA, USA) was applied to test Fe valence of Fe_7_(PO_4_)_6_ NMs and Avantage 5.9 (Thermo Fisher, Waltham, MA, USA) was used for data analysis. Zetasizer (Nano-ZS90, Malvern Instruments, Malvern, UK) was used to examine the surface charge (zeta potential) and hydrodynamic diameter (Dh) of the Fe_7_(PO4)_6_ NM suspension.

### 2.2. Plant Cultivation and NM Exposure

Tomato seeds (*Solanum lycopersicum* L., No. 1 Jinpeng, Xi’an Jinpeng seed Co., Ltd., Xian, China) were chosen with similar size, sterilized with 5% NaClO for 10 min, and thoroughly washed with DI water three times. The sterilized seeds were soaked in DI water for 4 h and then laid flat on totally wet filter paper and incubated in darkness at 25 °C for 48 h to germinate. After that, the tomato seedlings of uniform size and degree of germination were transferred to soil homogenized with Fe_7_(PO_4_)_6_ NMs in pots for seedling and fruit experiments.

Fe_7_(PO_4_)_6_ NMs were mixed with soil vigorously to guarantee that soil was homogenized with the NMs thoroughly before tomato seedling cultivation. The soil was collected from a farm located in Wuxi (latitude 31.52° N and longitude 120.13° E, China) and the properties of soil are shown in Appendix A. 600 g amended soil in each pot was used in seedling experiment and the plants were harvest after 42 days of cultivation. 4 kg amended soil in each pot was used in a fruit experiment and tomato fruits were collected after 115 days cultivation. Wang et al. (2020) [29] concluded that the impacts of NMs on crop growth exhibited a trend of low-promoting and high-inhibiting. Fe-based NMs shows increasing positive effects trend with the concentration (<50 ppm), but positive trend decreased if NMs concentration was more than 50 ppm. Thus, 50 mg kg^−1^ of Fe_7_(PO_4_)_6_ NMs was selected for enhancing plant growth, and 5 ppm was chosen to be set as comparison to 50 ppm.

The plants were grown under greenhouse conditions, with temperatures of 28/20 °C for day and night, and a relative humidity of 60%. Each pot was set as a biological replicate, and each treatment contained six replicates. In the seedling experiment, each pot retained two seedlings to gain enough biomass for tests. As for fruit experiments, one seedling was cultured in each pot. 5 and 50 mg kg^−1^ were set in a preliminary experiment to explore their impacts on flavonoid accumulation. The results indicated that 50 mg kg^−1^ Fe_7_(PO_4_)_6_ NMs better promoted flavonoid content in fruits; therefore, 50 mg kg^−1^ Fe_7_(PO_4_)_6_ NMs was used in the following experiments. Fe-EDTA considered as commercial Fe fertilizer, which contained equivalent mass of Fe (20.4 mg kg^−1^) with 50 mg kg^−1^ Fe_7_(PO_4_)_6_ NMs, was mixed in the same way as NMs and was also conducted in the seedling experiments. “CK” was defined as control.

### 2.3. Photosynthesis, Root Parameters, Element Content, and Single Particle Concentration

The relative chlorophyll content of tomato leaves was measured by Chlorophyll Meter (SPAD-502 Plus, Konica Minolta Inc., Tokyo, Japan) when the tomato grew to the fourth leaf stage (the fourth true leaf was fully unfolded). Net photosynthetic rate (Pn), stomatal conductance (Gs), transpiration rate (E), and intercellular CO_2_ concentration (Ci) were measured by CIRAS-3 portable gas exchange system (CIRAS-3, PP-Systems, Amesbury, MA, USA). The root morphology was scanned by WinRHIZO Pro 2017 b (Regent Instruments Inc., Ville de Québec, Quebec, Canada).

For the determination of nutrient content, 25 mg oven-dried tomato shoots and roots of fourth leaf stage (six replicates) were added into a digestion tube with 3 mL ultra-pure water and 3 mL nitric acid successively, and put into the digestion apparatus (MARS 6, CEM, Matthews, NC, USA), then the digestion was carried out at 190 °C, 1400 W. After digestion, the liquid was filtered by a 0.22 μm filter and diluted to 50 mL. The protocol of sample treatments was based on Wang et at. (2021) [30] with modifications. 25 mg of the tissue in fourth leaf stage was washed three times with DI water and then homogenized in 3 mL 20 mM 2-(*N*-morpholino) ethanesulfonic acid (MES) buffer (pH = 5.0). Then, the mixture was shaken at 37 °C for 24 h. After centrifugation at 12,000 rpm for 5 min, the supernatant was collected and taken through 0.45 μm filter membrane. The leachate was double diluted with DI water before single particle test. The content of Fe_7_(PO_4_)_6_ NMs in plant shoots, roots, and stems was determined by single particle ICP-MS (SP-ICP-MS, Thermo Fisher, Bremen, Germany) and parameters setting of testing were referred to Xu et al. (2020) [31].

### 2.4. Net H^+^ Flow Rate, IAA Content and Flavonoid Content

The protocol of net H^+^ flow rate determination was based on Ye et al. (2021) [32] and performed on Non-invasive Micro-test Technology System (NMT, 100S-SIM-XY, Xuyue Sci & Tech Co., Ltd., Beijing, China). The H^+^ microsensors were prepared as follows: a microsensor (Φ4.5 ± 0.5 μm, XY-CGQ-01, Xuyue, China) was placed 1 cm from the tip and filled with H^+^ exchange liquid (15 mM NaCl, 40 mM KH_2_PO_4_, pH = 7.0). Then, the microsensor was filled with H^+^ exchange reagent (LIXs, XY-SJ-H-10, Xuyue Sci & Tech Co., Ltd., Beijing, China). Before measuring H^+^ flow, the microsensors were calibrated with H^+^ calibration solutions (0.1 mM CaCl_2_, pH = 5.5 and pH = 6.5) and kept Nernst slope at 58 ± 5 mV decade^−1^. The tomato seedlings of fourth leaf stage were washed with DI water and the roots were completely immersed in test solution (0.1 mM CaCl_2_, 0.3 mM 2-(*N*-morpholino) ethanesulfonic acid, pH = 6) for 30 min of equilibration. The intact roots were selected on the Non-invasive Micro-test Technology System (NMT, 100S-SIM-XY, Xuyue Sci & Tech Co., Ltd., Beijing, China). The intact and unbroken roots were selected under the microscope and initial position of microsensor tip was set at 5 μm from the root surface. The root elongation region was selected as the probe detection region [33,34,35]. The step size of the probe was set to 20 μm, and the H^+^ flow rate was collected every 4 s. The data processing was performed by imFluxes V2.2 software.

IAA content determination was performed on Ultra performance liquid chromatography coupled with electrospray ionization and mass spectrometry (UPLC-ESI-MS/MS) based on the studies of Xiao et al. (2019) [36]. In brief, 100 mg from fourth leaf stage was ground in liquid nitrogen and added with 1 mL ethyl acetate, including 10 μg mL^−1^ butylated hydroxytoluene. After vortexing for 15 min, the samples were ultrasonically extracted in ice for 15 min. After centrifugation at 4 °C, 12,000 rpm for 10 min, the supernatant was transferred into a new tube and gently evaporated to dryness under nitrogen flow. The residual was redissolved in 200 μL 70% ethanol (*v*/*v*), 10 μL of the solution was directly injected in UPLC-ESI-MS/MS. UPLC-ESI-MS/MS equipment system setup was described in Appendix A. The methods of flavonoid extraction and detection were described in Appendix A.

### 2.5. Sucrose Content in Fruits

The method of sucrose content in tomato fruits was measured based on Feng et al. (2019) with slight modifications [37]. Dried samples were homogenized with 3 mL of 80% (*v*/*v*) ethanol and heated to 80 °C in water bath for 10 min; the sediment was collected by centrifugation at 8000 rpm for 10 min. The extraction process was carried out another three times as described above, and supernatant was collected each time. Next, 10 mg activated carbon was added to supernatant for decoloring and 0.1 mL of 0.1 M NaOH solution was added into 0.9 mL of supernatant. 3 mL of 10 M HCl and 1 mL of 0.1% resorcinol solution (0.1 g of resorcinol with 100 mL of 95% ethanol) were added to the supernatant in 80 °C water bath for 30 min. After the supernatant cooling down to room temperature, the sucrose concentration was determined using a microplate reader (Thermo Scientific, Waltham, MA, USA) at 480 nm.

### 2.6. Quantitative Real-Time PCR (qRT–PCR)

Gene expressions of flavonoid synthesis related genes (*SlPAL*, *SlC4H*, *Sl4CL*, and *SlCHI*) in fruits and PM H^+^ ATPase (*LHA1*, *LHA2*, and *LHA4*) in roots were determined. Sucrose transporter related genes of *SlSUT1*, *SlSUT2*, and *SlSUT4* were examined in seedling leaves. The sequences of primers of internal reference gene and other target ones were listed in Appendix A. The plant samples were ground into powder in liquid nitrogen. According to plant RNA extraction protocol (TaKaRa MiniBEST Plant RNA Extraction Kit), total RNA of roots and fruits was isolated, which was used as templates to proceed reverse transcription using T100 Thermal Cycler (Bio-Rad, Hercules, CA, USA). The reverse transcription procedure was carried out with EasyQuick RT MasterMix (CWBIO) kit and the conditions was as follow: 37 °C for 15 min, 85 °C for 5 s, and stored at 4 °C to obtain cDNA for PCR (polymerase chain reaction). Real-time PCR was executed with UltraSYBR Mixture (CWBIO) as fluorochrome operating on a CFX96 Real-Time system combined with C1000 Touch Thermal Cycler (Bio-Rad, Hercules, CA, USA). The targeted DNA amplification procedure was set as follows: (1) pre-degeneration at 95 °C for 10 min, (2) denaturation at 95 °C for 15 s, followed by (3) 60 °C for 1 min, and a total of 40 cycles of (2) to (3) were performed. The standard calculation 2^−ΔΔCT^ was used to measure the relative expression of the targeted genes [38].

### 2.7. Transcriptomic and Metabolomic Analysis of Tomato Fruits

Transcriptomic analysis of tomato fruits was performed by Genedenovo Biotechnology Co., Ltd. (Guangzhou, China). Briefly, the expression level of each gene was measured by fragments per kilobases per millionreads (FPKM). The differences in RNA expressions were analyzed by DESeq2, DEGseq and edgeR. The systematic analysis of gene function was conducted by comparison to the Kyoto Encyclopedia of Genes and Genomes (KEGG) and Gene ontology (GO) databases.

The extraction, identification, and quantification of tomato metabolites were conducted according to García et al. (2017) [39]. A one hundred microgram fruit sample was put in 2 mL tubes, and was mixed with 1.5 mL methanol/water (80:20, *v*/*v*) containing 0.2 mg L^−1^ 2-chloro-L-phenylalanine as internal standard. Subsequently, the mixture was vortexed vigorously and ultrasonicated for 30 min in ice, then centrifuged at 4 °C 12,000 rpm for 15 min. The supernatant was analyzed by UPLC-ESI-MS/MS (Thermo Fisher, Germering, Germany). The detailed condition of UPLC-ESI-MS/MS and data analysis is shown in Appendix A. The metabolic data processing is demonstrated in Appendix A.

### 2.8. Statistical Analysis

All data were analyzed using SPSS (IBM SPSS Statistics 25) [40] by one-way ANOVA with LSD and T-test. The threshold of significant difference of results was at *p* < 0.05 between controls and treatments. The data were reported as the mean and standard deviation (*n* = 6). Different letters represent significance of *p* < 0.05 and asterisk quantity represent significance as the following: “*” for 0.01 < *p* < 0.05, “**” for 0.001 < *p* < 0.01 and “***” for *p* < 0.001.

## 3. Results and Discussion

### 3.1. Fe_7_(PO_4_)_6_ NMs Characterization

The TEM image (Figure 1a) showed that the shape of Fe_7_(PO_4_)_6_ NMs was rod-like and 60 nm in width. EDS spectra (Figure 1b) demonstrated that the NMs were composed of Fe, P, and O. XRD patterns (Figure 1c) exhibited the characteristic peaks of the NMs consistent with the peaks annotated on the standard Fe_7_(PO_4_)_6_ card (PDF Card No. 72-2446). The XPS results (Figure 1d) displayed two prominent peaks at 712.5 eV and 725.9 eV, corresponding to Fe 2p_3/2_ and Fe 2p_1/2_, respectively. Fe^3+^ and Fe^2+^ formed Fe 2p_3/2_ peaks at 712.06 eV and 717.24 eV, respectively. Fe 2p_1/2_ also had peaks at 725.35 eV (Fe^2+^) and 730.45 eV (Fe^3+^). The Fe^2+^ in Fe_7_(PO_4_)_6_ NMs might be caused by the partial reduction of Fe^3+^ in FePO_4_ because of the presence of carbon from PVP K-30 [41]. The hydrodynamic diameter and Zeta potential of the NMs were 716.89 ± 60.32 nm and −14.37 ± 1.09 mV displayed as Appendix A.

### 3.2. Fe_7_(PO_4_)_6_ NMs Enhanced the Flavonoids Accumulation in Tomato Fruits and Related Gene Expressions

After 115 days’ growth, the amendment of Fe_7_(PO_4_)_6_ NMs with 5 and 50 mg kg^−1^ had no significant effect on the total fruit weight of tomatoes (Figure 2a,b). However, the flavonoid content in tomato fruits was significantly enhanced by the amendment of Fe_7_(PO_4_)_6_ NMs with 5 and 50 mg kg^−1^ (Figure 2c–e). It was reported that naringenin, quercetin and rutin were three kinds of main flavonoids in tomato fruits [42], Besides, quercetin and rutin belong to flavonol (a subclass of flavonoids) and the latter is a vital downstream metabolite of the former [42,43]. More importantly, flavonol contains a hydroxyl group at the C3 position, which makes flavonols excellent antioxidants in tomatoes [43]. The content of naringenin was significantly enhanced by 58.54% and 681.39% after the amendment of 5 mg kg^−1^ and 50 mg kg^−1^ Fe_7_(PO_4_)_6_ NMs as compared with control, respectively (Figure 2c). The content of quercetin and rutin in tomato fruits was increased by 33.07 and 80.20% upon 5 mg kg^−1^ Fe_7_(PO_4_)_6_ NMs exposure, and increased by 54.59 and 120.36% upon 50 mg kg^−1^ Fe_7_(PO_4_)_6_ NMs exposure as compared with control, respectively (Figure 2d,e).

Flavonoids are derived from the L-phenylalanine via the general phenylpropanoid pathway [44]. The naringin is produced from L-phenylalanine by a series of enzyme catalyzation, which are encoded by *SlPAL*, *SlC4H*, *Sl4CL*, *SlCHS1*, and *SlCHI* [44]. The relative expression of *SlPAL*, *SlC4H*, *Sl4CL*, *SlCHS1*, and *SlCHI* was significantly upregulated by 83.54, 90.83, 189.09, 24.16 and 105.92% after amendment with Fe_7_(PO_4_)_6_ NMs, respectively (Figure 2f). Meanwhile, *SlF3H*, *SlF3′H*, *SlFLS*, and *Sl3GT*, which encode the enzyme catalyzing naringenin to quercetin and rutin [45,46,47], were significantly upregulated by 54.76, 189.07, 94.73, and 96.15% in Fe_7_(PO_4_)_6_ NMs treatment as compared with control, respectively (Figure 2f). Flavonoids in plants is highly regulated by sucrose, which can act as a signaling molecule to alter genes expression in the flavonoids biosynthetic pathway [12]. Moreover, sucrose in tomato fruits mainly comes from photosynthetic system which requires the participation of various elements. PM H^+^-ATPase pumps proton out of cell to achieve acidification and facilitate the transport of various nutrients. The possible mechanism of flavonoids accumulation in the present study related to PM H^+^-ATPase activity, nutrient uptake, photosynthesis, and sucrose accumulation will be discussed in the following sections.

### 3.3. Fe_7_(PO_4_)_6_ NMs Improved the Growth of Tomato Seedlings

Visible growth promotion was observed in tomato seedlings upon Fe_7_(PO_4_)_6_ NMs exposure based on the phenotypic images (Figure 3a). As compared with equivalent Fe mass Fe-EDTA treatment and control, 50 mg kg^−1^ Fe_7_(PO_4_)_6_ NMs significantly enhanced tomato root fresh weight by 25.97 and 83.02%, respectively (Appendix A). Tomato shoot fresh weight was also enhanced by 16.67 and 47.78% in 50 mg kg^−1^ Fe_7_(PO_4_)_6_ NMs treatment as compared with equivalent Fe mass Fe-EDTA treatment and control, respectively (Appendix A). The amendment with 50 mg kg^−1^ Fe_7_(PO_4_)_6_ NMs significantly enhanced the root tip numbers by 262.83 and 215.84% as compared with Fe-EDTA and control (Figure 3b). The length, surface area, and volume of roots also showed a significant increase upon Fe_7_(PO_4_)_6_ NM application (Appendix A). The relative content of chlorophyll (SPAD) in tomato leaves was significantly increased by 7.39 and 36.32% upon Fe_7_(PO_4_)_6_ NMs amendment as compared with Fe-EDTA and control, respectively (Appendix A). At the same time, Fe_7_(PO_4_)_6_ NMs elevated Pn by 14.81 and 31.45% over Fe-EDTA and Control (Figure 3c). Gs, Ci, and E were also more improved by 19.30, 0.74 and 4.45% than Fe-EDTA, and by 39.05, 3.79 and 19.69% than control, respectively (Appendix A). The increase of SPAD value and photosynthetic parameters fully indicated that the application of Fe_7_(PO_4_)_6_ NMs significantly improved the photosynthesis and growth potential of tomato seedlings. It is reported that Fe-based NMs amendment could enhance the nutrient element uptake, and then promote the photosynthesis and growth of plants, as Fe is a crucial element for photosynthesis [48,49]. Therefore, the content of nutrient element in tomato shoots and roots was analyzed.

In roots, Mn content in NMs treatment was markedly more accumulated by 38.75% than Fe-EDTA and 54.11% than control (Figure 3d). Fe content in Fe_7_(PO_4_)_6_ NMs was increased by 31.85 and 42.86% over Fe-EDTA and control, respectively (Figure 3d). In shoots, Fe content greatly increased by 69.74 and 77.01% over Fe-EDTA and control. Mn content significantly rose by 82.89 and 80.26% over Fe-EDTA and control. In addition, Mg accumulated significantly by 35.99 and 31.60% over Fe-EDTA and control, respectively (Figure 3e). Fe contributes to electron transfer process, and is essential for cytochrome P450 [50]. Mn is a key element for oxygen evolution required in OEC (oxygen evolving complex) existed in PSⅡ (Photosystem Ⅱ) [51]. Mg acts as cofactor of a series of enzymes involved in photosynthesis carbon sequestration and metabolism [52]. The increase of Fe, Mn and Mg surely strengthened photosynthesis. In addition, during the flavonoid synthesis, Fe was vital for 2-ODD oxygenases in flavonoid synthase [26]. The obvious accumulation of Fe in tomato leaves can also elevate flavonoids synthesis potential. The enhanced Fe content in tomato seedlings could be attributed to (1) Fe_7_(PO_4_)_6_ NMs as an efficient Fe source, which enhanced the bioavailable Fe content in soils. It was reported that FePO_4_ NMs applied in soil could act as an efficient source of Fe and P as compared to bulk FePO_4_ due to its sub-micron size, which make it easier to reach the root surface and rapidly dissolved [53]. Our results of SP-ICP-MS demonstrated that Fe-bearing particles (average 288.62 nm) were detected in roots upon Fe_7_(PO_4_)_6_ NMs exposure (Figure 3f) and might explain how Fe_7_(PO_4_)_6_ NMs entered the roots and enhanced bioavailable Fe in tomatoes; (2) Fe_7_(PO_4_)_6_ NMs activated the response of tomato to acidify the rhizosphere soils, subsequently enhancing the bioavailability of Fe.

### 3.4. Fe_7_(PO_4_)_6_ NMs Promoted PM H^+^ ATPase and IAA Content in Root

Based on NMT results, H^+^ in the root elongation region showed extremely significant outflow, at 5.87 pmol·cm^−2^·s^−1^ after the application with Fe_7_(PO_4_)_6_ NMs (Figure 4a). With the treatment of Fe-EDTA, roots showed little net H^+^ efflux (0.37 pmol·cm^−2^·s^−1^) which meant effect of rhizosphere acidification was weaker compared to control (−0.60 pmol·cm^−2^·s^−1^) (Figure 4a). The relative expressions of *LHA* in roots were examined and results showed that *LHA2* and *LHA4* were significantly upregulated with Fe-EDTA by 1.74 and 4.44 folds compared to control, respectively, when no significance of *LHA1* was observed (Figure 4b). NMs significantly upregulated *LHA1*, *LHA2* and *LHA4* by 3.28, 1.30 and 8.94 folds over control, respectively (Figure 4b). IAA plays a crucial role in activating plasma membrane (PM) H^+^-ATPase [54]. The concentration of IAA in Fe-EDTA was increased by 66.67% but with no significance over control. Fe_7_(PO_4_)_6_ NMs treatment significantly increased IAA content by 70.75 and 164.21% as compared to Fe-EDTA and control, respectively (Figure 4c). The above results indicated that NMs induced more IAA accumulation and significantly activated more PM H^+^-ATPases to pump more proton and acidify rhizosphere to obtain nutrient (Figure 4) [55]. The ensuing decrease of apoplastic pH would alter the activity of modification proteins, e.g., pectin methylesterases, xyloglucan endotransglycosylase/hydrolases, and expansins, leading to the changes in extensibility of root cell wall. These results illustrated that the tomato roots were stimulated to pump large number of protons to acidify the rhizosphere by Fe_7_(PO_4_)_6_ NMs. A few researches mentioned that NMs activated PM H^+^ ATPase in roots and pump out protons into apoplast and acidify their rhizosphere, finally increased NH_4_^+^, P, K, Fe availability in soil [23,24]. According to acid growth theory [56], the decrease of apoplast pH activates cell wall-loosening enzymes, which could make cell expand along with turgor pressure and increase root elongation. H^+^ flux and PM H^+^ ATPase activity also has critical influence on elements uptake of roots. Plants have a strategy that releases protons into the rhizosphere by PM H^+^ ATPase, reducing the pH of the rhizosphere to induce elements dissolution, thus increasing the absorption of nutrients [20]. Moreover, elevated PM H^+^-ATPase activity could hyperpolarize the PM, increasing the energy for nutrient uptake, which is necessary to maintain water uptake and thus the turgor pressure that forces cell wall expansion.

### 3.5. Fe_7_(PO_4_)_6_ NMs Up-Regulated Sucrose Transporter and Sucrose Accumulation in Fruits

Sucrose is the product of photosynthesis in leaves and is transferred to other tissues by SUT (sucrose transporter) [57]. To further investigate the in planta sucrose transport, gene expressions of sucrose transporter in leaves of fourth leaf stage were analyzed. The qRT-PCR results demonstrated that *SlSUT1*, *SlSUT2*, and *SlSUT4* in plants exposed to Fe_7_(PO_4_)_6_ NMs were upregulated significantly over control and Fe-EDTA (Figure 5a). The up-regulation of *SlSUT* in leaves indicated sucrose was transported from leaves to flowers more efficiently; tomato fruit setting and fruit sucrose accumulation would be thus increased [19]. Accordingly, sucrose concentration in tomato fruits by Fe_7_(PO_4_)_6_ NMs was significantly increased by 21.15% as compared with control (Figure 5b). Sucrose upregulates flavonoid biosynthetic genes and induces expression of transcription factors to promote flavonoid synthesis in fruits [14]. It was reported that sucrose acted as an endogenous inducer in upregulating the expression of anthocyanin biosynthetic and regulatory genes of *MrMYB1* [58]. Sucrose would also increase transcript levels of early anthocyanin biosynthesis genes including *CHI*, *CHS*, and *C4H* in Arabidopsis [59,60]. Also, sucrose could be metabolized into glucose and the latter acted as the substrate of pentose phosphate pathway for providing flavonoid synthesis precursors [11]. As sucrose accumulation increased in NM treatment, it could act as a signal to upregulate significantly flavonoid synthesis genes and might provide more precursors in tomato fruits compared to control.

### 3.6. Mechanisms of Enhanced Flavonoids Accumulation in Tomato Fruits by Integration Analysis of Transcriptomics and Metabolomics

#### 3.6.1. Transcriptomic Analysis of Tomato Fruits by Fe_7_(PO_4_)_6_ NMs

To explore the molecular mechanism underlying the enhanced flavonoid synthesis by Fe_7_(PO_4_)_6_ NMs, transcriptomic analysis of tomato fruits was performed. PCA (principal component analysis) plot showed that CK and Fe_7_(PO_4_)_6_ NMs treatment were clearly separated along PC1 (92% of total variance) (Figure 6a). The differentially expressed genes (DEGs) were annotated and used for GO (Figure 6b and Figure 7a) and KEGG (Figure 6c and Figure 7b) enrichment analysis. For the functional classification in GO database, DEGs annotated with “cellular process”, “metabolic process”, “biological process” and “response to stimulus” were most abundant within the biological process category (Appendix A). In molecular function, DEGs were enriched in “catalytic activity” and “binding”. As for cellular component, DEGs were enriched in “cell part”, “organelle” and “membrane”. “metabolic process”, “response to stimulus” and “catalytic activity” were related with flavonoid synthesis [61,62], and secondary metabolic process, phenylpropanoid metabolic process, and secondary metabolite biosynthetic process were significantly altered upon Fe_7_(PO_4_)_6_ NMs relative to CK. In order to further understand the DEG functions, the KEGG pathway enrichment was performed. Among analysis results, pathways related to flavonoid synthesis like phenylpropanoid biosynthesis (ko00940), biosynthesis of secondary metabolites (ko01110) and phenylalanine metabolism (ko00360) were significantly enriched in tomato fruits (Figure 6c and Figure 7b).

Specifically, there were 11 DEGs of tomato fruits involved in flavonoid synthesis (Appendix A). The transcriptional level of *SlSS*, *SlTPS6*, *SlTPS11* and *SlTKL-1*, belonging to sucrose metabolism pathway [63,64,65], was significantly upregulated by 101.43, 131.24, 118.71 and 32.17% over CK, indicating that Fe_7_(PO_4_)_6_ NMs elevated transcriptional regulation of sucrose metabolism pathway. In shikimic acid pathway, only *SlDHQ-SOR* was detected, and was significantly downregulated by 15.20%; thus, shikimic acid pathway was suppressed by Fe_7_(PO_4_)_6_ NMs. Li et al. [66]. found that most of the shikimate pathway genes showed a decreased transcript abundance in strawberry fruits injected with 100 μL of ABA (1 μM). Our transcriptomic results showed that *PYL*, the gene encoding ABA receptor, was significantly upregulated (188.46%) with Fe_7_(PO_4_)_6_ NMs treatment. High expression of *PYL* represented ABA receptor was activated, indicating that ABA content was increased [67,68]. In phenylalanine pathway, transcriptional level of *SlEPSPS-1* (encoding the enzyme catalyzes the formation of EPSP (5-enolpyruvyl shikimate-3-phosphate) from shikimate-3-phosphate and phosphoenolpyruvate) [69] and *SlCM2* (encoding the enzyme rearranges the enolpyruvyl side chain of chorismate to form prephenate) [70] were significantly downregulated by 15.63% and 34.99% (Appendix A). In final flavonoid biosynthesis, the transcriptional level of *Sl4CL* and *SlCHI* was significantly increased by 20.75 and 123.53%. In tomato fruits, *SlMYB12* was examined as the TF correlated well with the expression of flavonoid synthesis genes [71,72,73]. SlMYB12 can lead to the flavonoid accumulation and strongly regulate flavonoid synthesis genes [73], and was upregulated significantly by Fe_7_(PO_4_)_6_ NMs (Appendix A). Overall, the expression pattern of genes in fruits upon Fe_7_(PO_4_)_6_ NMs showed a preference for final flavonoid biosynthesis activation.

#### 3.6.2. Metabolic Analysis of Tomato Fruits Exposed to Fe_7_(PO_4_)_6_ NMs

The impact of Fe_7_(PO_4_)_6_ NMs on metabolic profiles of tomato fruits was further investigated; the PCA score plot (Appendix A) indicated that metabolic levels were significantly different between CK and Fe_7_(PO_4_)_6_ NMs (66.3% of total variance). In all samples, 217 metabolites were identified and annotated and 50 relative quantitation of differentially expressed metabolites (DEMs) were selected according to the standard of VIP (variable importance in projection) > 1 and *p* < 0.05 and illustrated (Appendix A). In sucrose metabolism pathway, UDP-glucose (the production of sucrose catalyzed by sucrose synthase), trehalose and D-Glucose-6P (G6P) were detected and increased to 17.50, 28.38 and 10.13% compared with control, respectively (Appendix A), and it was because the transcriptional levels of *SlSS*, *SlTPS6* and *SlTPS11*, regulating enzymes catalyzing sucrose to trehalose, were significantly upregulated (Appendix A) [74,75]. However, the metabolites of shikimic acid pathway and phenylalanine synthesis pathway were not detected.

In this study, 11 DEGs and 6 DEMs were identified in the process of sucrose metabolism to the final flavonoid biosynthesis of tomato fruits. Three transcriptional genes (*SlSS*, *SlTPS6* and *SlTPP*) of enzymes catalyzing sucrose to trehalose were significantly upregulated by Fe_7_(PO_4_)_6_ NMs in tomato fruits (Figure 8a) [63,64,65]. The significant increase of UDP-glucose (product of sucrose degradation regulated by *SlSS*) markedly elevated the content of trehalose (product of the catalytic processes regulated by *SlTPS6* and *SlTPP*). These up-regulations of DEGs and DEMs could jointly promote the significant increase of G6P (Appendix A), which is a key common precursor of glycolysis and pentose phosphate process, and its final products PEP and erythrose-4-phosphate (E4P) serve as raw materials for shikimic acid pathway [76]. In addition, although no DEG was detected in the pentose phosphate pathway, *SlTKL-1*, which regulates the production of E4P derived from G6P in glycolysis, was significantly increased. Therefore, Fe_7_(PO_4_)_6_ NMs significantly increased the transcriptional regulation level of sucrose metabolism in tomato fruits and accelerated sucrose conversion, providing more adequate materials for downstream pathways.

Surprisingly, *SlDHQ-SOR* detected in the shikimic acid pathway was significantly down-regulated (Figure 8b). Increased ABA significantly down-regulated the shikimic acid pathway related genes during strawberry ripening [66]. We investigated the transcription level of ABA signaling pathway and found that *SlPYL4*, the regulation gene of ABA receptor PYL was significantly elevated, suggesting that ABA response pathway was activated in tomato fruits and inhibited the expression of *SlDHQ-SOR* in shikimic acid pathway [68]. ABA was obviously increased by NMs (Appendix A). Some studies demonstrated that ABA acts as a crucial signal to promote flavonoids [77,78]. In short, ABA enhances flavonoid synthesis, though shikimate pathway is suppressed. Thus, it could explain the reason why shikimate pathway is not consistent with increased flavonoid synthesis but regulated by endogenous ABA. The transcription level was also down-regulated in the phenylalanine synthesis pathway following shikimic acid pathway (Figure 8c), which may be caused by the decrease of substances flowing into phenylalanine synthesis pathway due to the decreased transcription level of a shikimic acid pathway.

In the final flavonoid biosynthesis pathway, two DEGs of *Sl4CL1* and *SlCHI* were significantly increased (Figure 8d). Sl4CL1 catalyzes 4-coumaric acid to *p*-coumaroyl-CoA, providing the precursor for the synthesis of naringenin chalcone. The transformation from naringenin chalcone to naringenin was catalyzed by SlCHI, and SlCHI was found to be the rate-limiting enzyme of flavonoid synthesis [79]. In our study, the upregulated *SlCHI* indicated that flavonoid biosynthesis was significantly accelerated. Most other types of flavonoids are modified based on naringenin [80]. The significantly increased content of naringenin (Figure 2c) illustrated enhanced downstream flavonoid level. Quercetin and rutin are the catalyzed products of naringenin in flavonol synthesis pathway. As the important antioxidant substances in tomato [81], the metabolic levels of these two flavonols were also significantly increased (Figure 2c), indicating that the antioxidant ability of tomato was significantly improved. Flavonoid synthesis is closely regulated by TFs such as *SlMYB12,* which was improved as an extensive positive role in flavonoid biosynthesis pathways of tomatoes [72,73]. The markedly upregulated transcriptional level of *SlMYB12* implied that flavonoid synthesis at the level of transcriptional regulation was enhanced by Fe_7_(PO_4_)_6_ NMs. Hence, Fe_7_(PO_4_)_6_ NMs promoted flavonoid synthesis by enhancing the transcription level and ultimately induced flavonoid accumulation in tomato fruits.

## 4. Conclusions

After treated with 50 mg kg^−1^ Fe_7_(PO_4_)_6_ NMs, the content of IAA in tomato seedlings roots was significantly increased, and the activity of PM H^+^ ATPase was enhanced over control and equivalent Fe mass Fe-EDTA fertilizer, leading to root development and rhizosphere acidification. The uptake of nutrient elements was elevated, subsequently enhancing the photosynthesis. The sucrose transporter genes of *SlSUT1*, *SlSUT2*, and *SlSUT4* were also markedly upregulated by Fe_7_(PO_4_)_6_ NMs, and more sucrose was transported from leaves to flowers to promote fruit setting and yield potential. Compared with the control, the accumulation of sucrose in fruit treated by Fe_7_(PO_4_)_6_ NMs indirectly led to the increase of flavonoid synthesis precursors. Transcriptome and metabolome data demonstrated the positive activation of a sucrose metabolism pathway and an ultimate flavonoid biosynthesis pathway in tomato fruits by Fe_7_(PO_4_)_6_ NMs, and qRT-PCR results verified the up-regulation of flavonoid synthesis genes. Our findings reveal the potential of NMs application to promote the quality of fruits with flavonoids, which is useful for the development of high-quality nano-enabled agriculture in the future.

## Figures and Tables

**Figure 1 nanomaterials-12-01341-f001:**
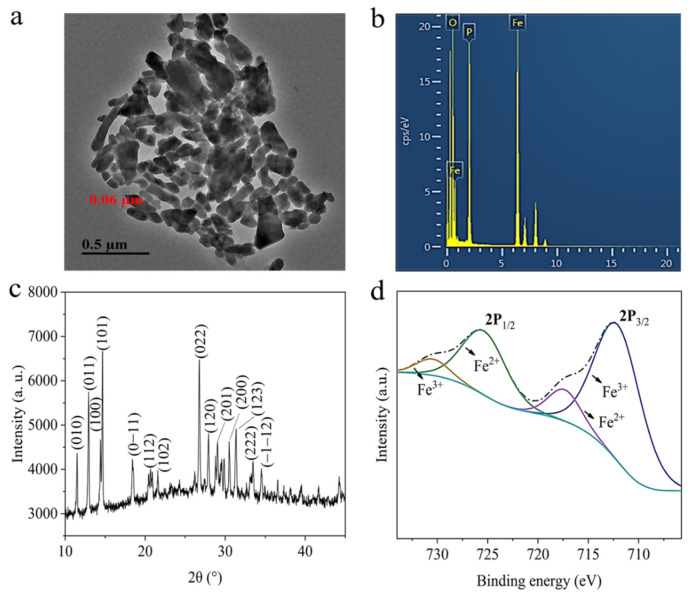
Characterizations of synthesized Fe_7_(PO_4_)_6_ NMs. (**a**) TEM image; (**b**) EDS image of Fe, P, and O elements; (**c**) XRD pattern; (**d**) XPS diagram.

**Figure 2 nanomaterials-12-01341-f002:**
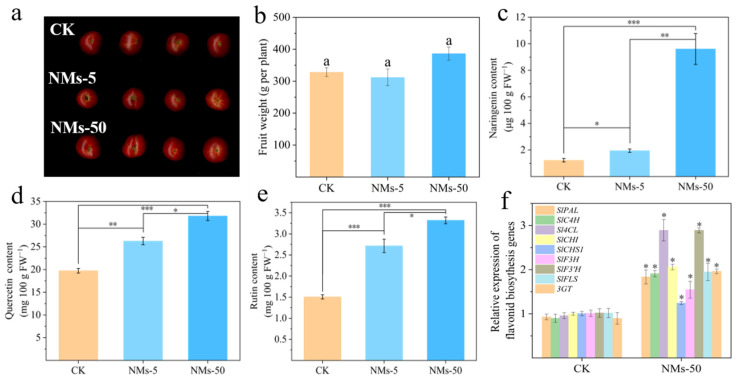
(**a**) Tomato fruits upon 5 and 50 mg kg^−1^ Fe_7_(PO_4_)_6_ NMs; (**b**) total fruit yield; (**c**–**e**) content of naringenin (**c**), quercetin (**d**) and rutin (**e**) in red mature fruits; (**f**), relative expression of flavonoids synthesis genes in tomato fruits. Letters of “a” in (**b**) represent no significance between treatments. Asterisk quantity represents significance as following: “*” for 0.01 < *p* < 0.05, “**” for 0.001 < *p* < 0.01 and “***” for *p* < 0.001.

**Figure 3 nanomaterials-12-01341-f003:**
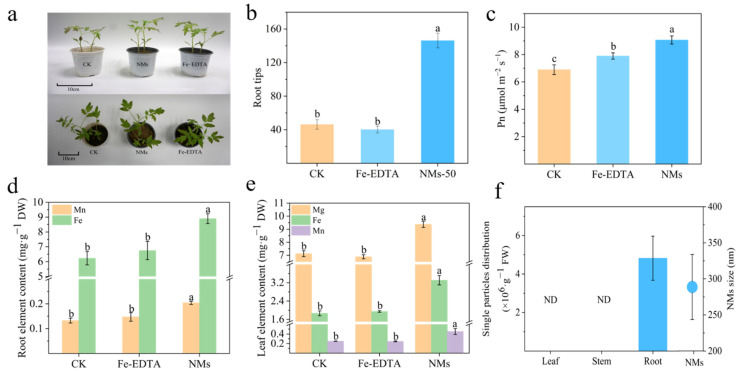
The impacts of Fe_7_(PO_4_)_6_ NMs on tomato seedling growth. (**a**) tomato phenotype; (**b**) root tips number; (**c**) Pn of the fourth fully unfolded true leaf; (**d**,**e**) element content of significant increased by NMs treatment in roots and shoots; (**f**) single particles distribution in leaf, stem and root of Fe_7_(PO_4_)_6_ NMs and average size of NMs. Different letters in (**b**–**e**) represent significance of *p* < 0.05. ND in (**f**) means not detected.

**Figure 4 nanomaterials-12-01341-f004:**
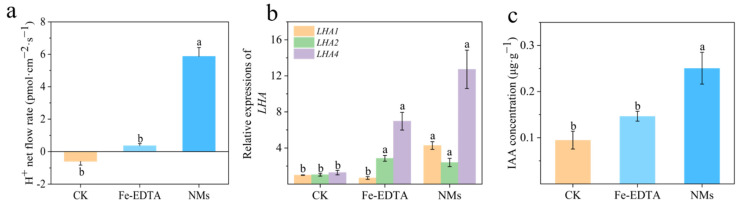
Fe_7_(PO_4_)_6_ NMs stimulated root PM H^+^ ATPase and IAA accumulation. (**a**) H^+^ net flow rate of tomato seedling root elongation zone. Negative value means H^+^ influx and positive value means efflux. (**b**) relative expressions of *LHA* in tomato roots. (**c**) IAA concentration in tomato roots. Different letters in represent significance of *p* < 0.05.

**Figure 5 nanomaterials-12-01341-f005:**
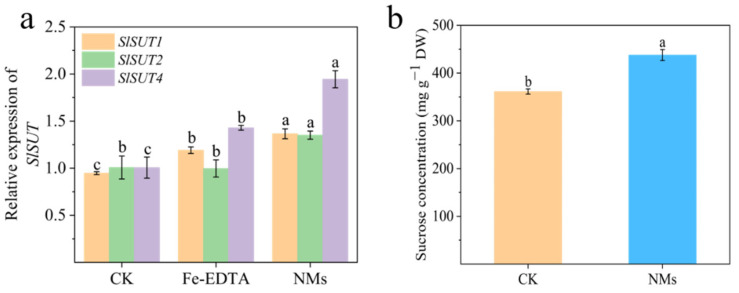
Relative expressions of sucrose transporter genes (**a**) and sucrose content in tomato fruits (**b**) Different letters in represent significance of *p* < 0.05.

**Figure 6 nanomaterials-12-01341-f006:**
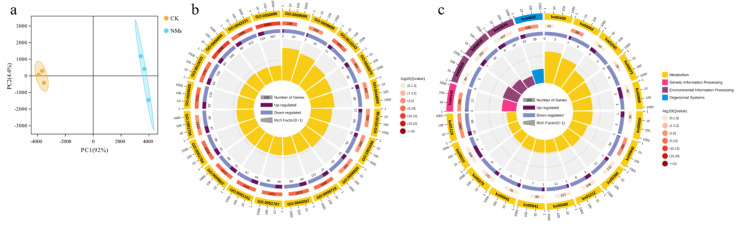
Transcriptomic profiling in tomato fruits upon Fe_7_(PO_4_)_6_ NM exposure. (**a**), PCA plot; (**b**), classification of DEGs in GO; (**c**), KEGG enrichment analysis of DEGs.

**Figure 7 nanomaterials-12-01341-f007:**
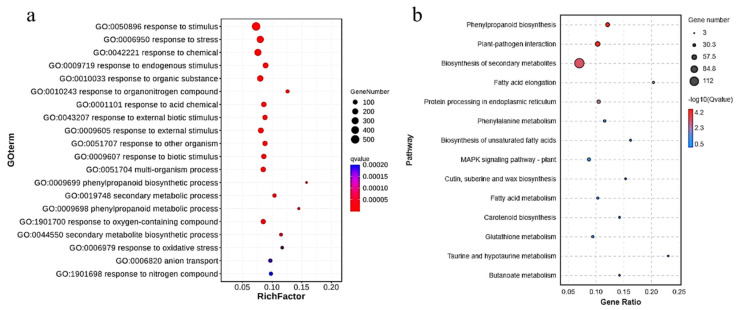
GO and KEGG enrichment of DEGs in tomato fruits upon Fe_7_(PO_4_)_6_ NMs exposure. (**a**) GO enrichment; (**b**) KEGG enrichment of pathways.

**Figure 8 nanomaterials-12-01341-f008:**
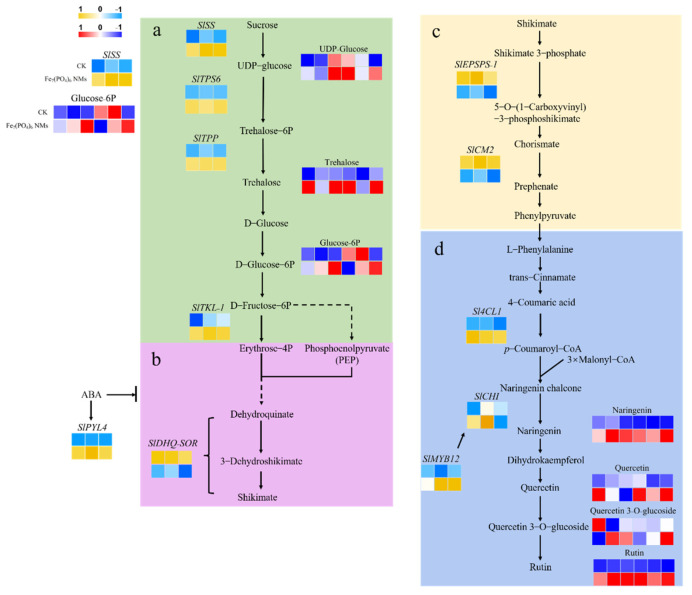
Transcriptomic and metabolomic pathways of flavonoids synthesis enhanced by Fe_7_(PO_4_)_6_ NMs in tomato fruits. (**a**) sucrose metabolism; (**b**) shikimic acid pathway; (**c**) phenylalanine synthesis pathway; (**d**) flavonoid biosynthesis pathway. The data of transcriptomics and metabolomics were normalized before comparison in heatmaps. The color scheme represents decrease or increase of value of log_2_(fold change) and the fold change indicated the FPKM ratio of DEG and response value ratio of DEM in 50 mg kg^−1^ NMs to control. At transcriptional level, blue represents downregulation and yellow means upregulation; at metabolic level, blue represents downregulation and red means upregulation. Upper squares represent samples of CK and lower ones represent samples of NMs.

## Data Availability

RNA-Seq data generated from *Solanum lycopersicum* L. has been submitted to the NCBI under project number PRJNA824998 (https://www.ncbi.nlm.nih.gov/sra/PRJNA824998, access date: 11 April 2022). The data presented in this study are available on request from the corresponding author.

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
