# Peer review of "Triiron Tetrairon Phosphate (Fe7(PO4)6) Nanomaterials Enhanced Flavonoid Accumulation in Tomato Fruits"

_nanomaterials, 2022, doi:10.3390/nano12081341_

Round 1

Reviewer 1 Report

The work done by Z. Wang and cols. Entitled “Triiron Tetrairon phosphate (Fe7(PO4)6) nanomaterials enhanced flavonoid accumulation in tomato fruits” in their study the authors made an analysis of amended soil with the synthetized Fe7 nanomaterial and the pathways involved in synthesis of tomato flavonoids under the presence of 5 or 50 mg of Fe7 nanomaterial. The study is well done, aims are covered by experimental strategies, and it is well written. However, the study presents some cavities, confusing data and some results are unclear.

  • In the manuscript authors did not explain the rationale to use two doses/amounts of Fe7 nanomaterial neither time of exposure to the compound. Please clarify in a statement.
  • Statistical analysis as are presented are unclear and make difficult to see the real effect of the nanomaterial. I strongly suggest to change standard deviation to standard error bars and make a clear statement of the meaning of symbols.
  • There are some typos and abbreviations no described
  • The introduction section stated on some transcription factors sucrose-activated. Please explain the role of those factors
  • It is unclear if Fe-EDTA results on figure 4 have some positive effects on the measured parameters. Please clarify.
  • How suitable in terms of costs is the use of Fe7 nanomaterial
  • How healthy could be to get more than 600% over production of naringenin in tomato with 50mg of Fe7 nanomaterial. Please discuss.
  • Please discuss the role of transcription factors sucrose-activated

Reviewer 2 Report

The manuscript by Wang et al. presents a well-designed study of the enhancement effect of (Fe7(PO4)6) nanomaterials on flavonoid accumulation in tomato. Using diverse approaches authors gave mechanistic explanation of observed effect. The strength of the study is the combined morphological, physiological and molecular methods in revealing mechanism of positive effect of (Fe7(PO4)6) nanomaterials on flavonoids. The results shown that Fe7(PO4)6 NMs promoted auxins, which enhanced root proton efflux and rhizosphere acidification leading to increased Mg, Fe, and Mn uptake into plants, subsequently, enhancing the photosynthesis. Transcriptome and metabolome data confirmed the positive activation of sucrose metabolism and up-regulation of flavonol synthesis genes. I believe the manuscript is appropriate for publication after addressing the following points:

  1. Please add more details about plant cultivations, when plants were transferred to soil, as seedlings? Were they planted in pots, how long the cultivation lasted? What was the composition of soil, pH of soil, watering of plant,s etc.?
  2. How was Fe7(PO4)6 NMs applied and when, for how long?
  3. What exactly were replicates? Pots or seedlings? It should be specified in more details.
  4. Were all analyses done in the same plant stage i.e. in fourth leaf stage?
  5. Part of the text should be checked for clarity.
  6. In figure captions please explain all abbreviations, as well as the meaning of letters and asterisks above columns. Also explain why did you sometimes use letters to mark significance and sometimes asterisks.
  7. It would be good to put results of content of nutrients important for photosynthesis and flavonoid biosynthesis that significantly change after NP exposure as main results not supplemental. The same goes for detection of Fe-bearing particles in roots as these finding are important.
  8. Some specific comments are marked in the text.

Reviewer 3 Report

The study presents some intriguing results, but there are many issues with the approach, presentation of methods and results, and it makes one question the evidence. The assumption is that increased Fe and P will increase flavonoids, and that using an NM form will be better than adding these elements separately. However, there is no information on the levels of Fe and P in the control, and the other treatment only considers Fe-EDTA, with apparently no additional P. Therefore, these are not valid comparisons. Key questions that the authors should address is: “what is the need to make the Fe7(PO4)6 into a nanomaterial? Why can’t it just be used directly?” Given that it takes a considerable amount of effort to make it into a nanomaterial, at least a treatment study with just exposure to Fe and PO4 fertilizer should be done.

While many advanced techniques are considered, there are several important missing details in the methods. In addition, the results don’t present all the treatments in all cases. Finally, there was no “optimization” of the concentration of NMs, since only two concentrations were considered.

In addition to increasing a few flavonoids, the only other major benefit was increased root and shoot biomass, which is not what a farmer is looking to sell.

The decrease in shikimate pathway is not consistent with increase flavonoid synthesis.

Abstract                                                              

Line 16 (L16) “provides the dawn” is an incorrect expression

L17 delete “the”

Introduction

L38 cardiovascular

L46 the sentence makes it sound like nanotechnology’s only purpose is to enhance flavonoid content in tomatoes. This is incorrect.

L53 is highly…  which can act

Methods

L99 What was the yield of the synthesis of the Fe7(PO4)6 NMs? Provide this in the results.

L118 Provide evidence that 50 mg kg-1 Fe7(PO4)6 NMs was an optimal concentration. Even though they were “preliminary experiments”, it is important to provide that information. Your study is incomplete without it. It seems you only did 5 and 50, which is insufficient to call this an optimization.

Section 2.2 what was the amount of fertilizer applied to the control, and in what form?

How were the Fe7(PO4)6 NMs added? Via the soil? How was it mixed? Or via the water solution? How did you enhance bioavailability of the NMs? Did you check to see that in fact Fe and P became more bioavailable to the plant?

Why was Fe-EDTA considered a reasonable alternative? The Fe is bound and not bioavailable. Explain.

L150 after centrifugation

L182 define “The standard calculation 2−ΔΔCT “ Not all readers are familiar with it.

Results

Figure 2 define CK. Why don’t you present Fe-EDTA results here?

Why do you think naringin was increased only in the 50 mg kg-1 application but not in the 5, while the other two flavonoids did have a response at 5?

L240 is highly

L257 most growers would not be interested in increased root & shoot weight. They sell the fruits, not the roots or shoots. In Figure 2 you clearly indicate that the mass of fruits did not change. The increase in photosynthesis may explain the increase in sugars.

L289-90 you mention sp-ICP-MS but you did not provide any details in Methods on this. How did you do the digestion of the roots? These may be Fe-bearing, but what is the composition?

L293 how did you determine that the NMs activated the response of tomato to acidify the rhizosphere soils? Sounds like speculation. Did you monitor soil pH?

Figures 3, 4 and 5 why do you only show NMs, and not the two different exposure concentrations?

Sucrose is a VERY common sugar, and thus its relationship to gene expression (e.g., anthocyanin biosynthesis, flavonoid synthesis) may be complicated. It would be better to look for other intermediate metabolites that would support the evidence that increased sucrose levels are responsible for the increased transcription.

L351 CK needs to be defined

L354 DEGs were classified, not enriched. Or if “enriched”, then provide an more quantitative analysis and indicate the statistical significance.

L368 is the analysis precise enough for 5 significant figures? What is the uncertainty associated with these measurements?

L394 how did you do the “relative quantitation”? How do you control for differential response of the LC-MS/MS for different classes of compounds? Were analytical standards used?

L398 without standards, how can you have such a high precision in the “increases”? What was your extraction efficiency for each of these metabolites? What methods were used to determine extraction efficiency?

Figure 7 the legends need to be explained in the caption, first the color scheme (fold times?) and then the different treatments or conditions being displayed. Not clear.

Figure 7d if the shikimate biosynthesis was downregulated, how does this result in enhanced flavonoid synthesis??

L438 TF needs to be defined

Conclusions

The decrease in shikimate pathway is not consistent with increase flavonoid synthesis.

Round 2

Reviewer 3 Report

Key questions that the authors should address is: “what is the need to make the Fe7(PO4)6 into a nanomaterial? Why can’t it just be used directly?” Given that it takes a considerable amount of effort to make it into a nanomaterial, at least a treatment study with just exposure to Fe and PO4 fertilizer should be done.
